# Are Public Subsidies to Encourage Young Farmers Effective? Case Study of a First-Time Farm Set Up by a Young Female Farmer in the Valencian Region of Spain

**Beatriz Llopis Gilabert \* and Isabel Pla-Julián \***

Applied Economics Department, University of Valencia, 46002 Valencia, Spain
\* Correspondence: beatriz.llopis@uv.es (B.L.G.); isabel.pla@uv.es (I.P.-J.)

**Abstract:** Generational renewal and the incorporation of women into the rural world are fundamental for the sustainability and modernisation of the agricultural sector. Hence the special government protection granted to the sector, which appears in both European legislation on rural issues and in the global strategy of the 2030 Agenda and the Sustainable Rural Development Goals involving a commitment to measures aimed at helping young farmers to set up agricultural holdings and especially at encouraging women to engage in farm management. In the case of Spain, this is nothing new, given that Law 35/2011 on the Shared Ownership of Agricultural Holdings became a veritable revolution in terms of gender in the rural world when it came into force. The results section discusses the practical application of funding for young farmers to start up an agricultural holding through a technical and economic study, and a case study of a first-time farm set up by a young female farmer. The main contribution of this work lies in answering two questions. The first one is to evaluate the efficiency of the subsidies for young people to set up farms in terms of offering a decent, stable livelihood for young men and women in Spain and the Valencian Region. The second one attempts to verify the effectiveness of both European and Spanish active public policies to encourae young men and women to join the agricultural sector.

**Keywords:** sustainability; young farmers; rural women; public subsidies; economic study; SDGs; farm; job opportunity

## 1. Introduction

There is a clear priority given to young farmers by providing them with funding for setting up farms. This is confirmed in current European, national, and regional legislation on rural issues, as well as in the global strategy of the 2030 Agenda and the Sustainable Rural Development Goals (SDGs) [1–4], specifically in the eighth goal. Furthermore, in line with the fifth SDG and the Law on the Shared Ownership of Agricultural Holdings [5–9], there is also a commitment to the empowerment of women, promoting their role in the management of agricultural holdings in order to achieve true equality between men and women in the rural world.

Factors such as an ageing agricultural workforce, the preference of young people for working in other economic sectors and the masculine nature of rural areas are the main reasons why the agricultural sector could be seriously compromised in the future and explains why its structure has not been modernised.

Precisely in order to ensure the sustainability of the agricultural sector, incentive strategies are needed to attract both young men and women to farming. In fact, if agriculture is perceived by young people as a job opportunity which can give them a decent livelihood, this could encourage young members of the working population to transfer from other productive industries to the agricultural sector. In this sense, aid to help young farmers set up business could have a direct impact on mitigating the high unemployment rates affecting this group, and especially affecting women. In fact, if we focus on women, the

effects of these measures could be manifold, considering that, as shown in subsequent sections, the Spanish labour market is characterised by dual segregation, both vertical and horizontal, with a poor female presence in the agricultural sector.

In this paper, we first study the regulatory background of public policies within the framework of measures relating to the access of young people and women to the agricultural sector. We examine the legal principles that gave birth to the rural development policy, the second cornerstone of the Community Agricultural Policy (CAP) through to the present day, including Spanish Law 19/1995 on the Modernisation of Agricultural Holdings and Spanish Law 35/2011 on the Shared Ownership of Agricultural Holdings, which is considered fundamental in the advancement of gender in the agricultural sector.

Subsequently, we analyse the specific situation of women in the agricultural sector and of young farmers in general, highlighting the poor visibility and outreach of female agricultural entrepreneurs, the problem of an ageing rural labour force with low numbers of young farmers, and men and women who prefer to work in other economic activities, such as the service sector.

Next, we focus on the legal framework for start-up funding for young farmers and we present a case study of a young female farmer setting up a first-time farm, verifying its economic viability, considering that the amount of this funding represents considerable external financing to defray the necessary start-up costs.

In addition, we will show that the positive discrimination given to young female farmers when scoring their funding applications meets the legislator's objective, which is to achieve real equality between men and women in the rural world, and thus complies with the principle of equality and non-discrimination set out in the Spanish Constitution [10]. This is seen as one of the most important challenges for the future of the agricultural sector.

Finally, we will examine the ineffectiveness in Spain and the Valencian Region of the support given to young farmers to set up a first-time farm and in general of the active public policies for this purpose, analysing the Spanish labour market and the current situation of women, as well as the number of male and female beneficiaries of funding during the current programming period of calls put out by the Valencian Regional Government.

## 2. Methodology

This research arises from field work based on an economic study of a farm, which determined the amount of money required to set up a farm and its economic viability based on the indicators established in Order 7/2015 of 1 December, which approves the terms and conditions for granting subsidies to young farmers to set up an agricultural holding [11].

The methodology described in this article centres on a case study of a first-time farm set up by a young female farmer. The case study method enables researchers to closely examine the data within a specific context, a small geographical area or a very limited number of individuals as the subjects of study, i.e., they examine data at micro level.

In short, we shall investigate a contemporary real-life phenomenon through a detailed contextual analysis of a limited number of events or conditions, and their relationships [12,13].

### 2.1. Farm Owner Profile

The profile of the farm owner is a woman between 18 and 40 years of age with university studies in economics, who is setting up a farm for the first time and for at least 10 years under individual ownership. This female owner must possess the professional qualifications and skills which are required to qualify for funding for young people to set up a first-time farm.

The chosen profile of the farm owner meets the two requirements of being young and female, which enables us to answer the question posed about the fact that support for first-time set up offers a decent, stable working future for young people in general, and women in particular.

## 2.2. Description of the Farm

The farm presented in this case study is located in the municipality of Xativa, in the province of Valencia. The geographical choice of the farm is basically due to two reasons:

Firstly, the municipality of Xativa is ideal for planting irrigated stone fruit crops, thus optimising resources. Secondly, we are linked to the municipality of Xativa because of an agricultural family tradition and because it is an area characterised by agricultural masculinization.

The land tenure regime is a leasehold with a total surface area of 11.61 hectares and the farm's end use is to grow fruit trees, namely stone fruit trees with irrigation. This means a total change of direction on the farm in order to improve profitability given that stone fruit trees with irrigation are more marketable.

The holding will be given priority status, as required by regulations, within 18 months and intends to start organic production within 24 months.

The farm includes a house with a well, which can be used by the farm's employee, and a warehouse for storing the machinery that will be acquired once the initial payment of the aforementioned subsidy has been obtained.

It should be noted that the deadline for implementing the investments proposed in the business plan is 36 months after the start-up date, making use, therefore, of the additional 18-month period established in Order 7/2015 to make these investments effective.

The justification for the use of this additional period lies in the need to have more time to make the planned investments effective, given that the farm is currently used to grow citrus fruit and 100% of the AWU is to be converted to stone fruit trees with irrigation.

In terms of sales, there are basically three options:

Firstly, retail distribution of the local product, i.e., it can be sold to retailers in the municipality of Xativa and the surrounding area of La Costera, to both specialised and large-scale retail outlets.

Secondly, online sales. An online service will be made available for dispatching orders placed through a website, guaranteeing the product's quality systems.

And finally, the farm will become a member of a cooperative to process and market at least 25 percent of its production. The cooperative is the Xativa Regional Agricultural Cooperative (COACXA COOP.V).

Therefore, this comprehensive presentation of the case study farm complies with the aforementioned requirements on applying for start-up funding for young farmers, as we will see later.

## 2.3. Annual Work Unit and Agricultural Income per AWU

In the technical and economic study of the farm, as we will see later, the annual work unit (AWU) was used as a reference indicator to calculate most of the costs required to start up a farm.

AWU is the full-time employment equivalent, i.e., one annual work unit corresponds to the work performed by one person on an agricultural holding on a full-time basis. The AWU is equivalent to 1920 h per year, i.e., the maximum number of working hours permitted under Spanish labour legislation.

To check the farm's economic viability, we compared agricultural income per AWU with the reference income fixed annually by the government for agriculture.

The agricultural income per AWU is obtained by

$$\frac{\text{(The net margin of the farm + The salaries paid)}}{\text{The number AWUs of labour force}}$$

The net margin of the farm: represents total gross margin main total overheads;
The salaries paid: represents total salaried workforce;
The number AWUs of labour force: corresponds to the AWU of the farm owner plus the AWUs of salaried labour.

*2.4. Efficiency and Effectiveness*

We used efficiency and effectiveness as the criteria to answer the two questions posed. Efficiency was the criterion used to answer the first question, which is whether public subsidies for first-time farm set up contribute to the employability of young men and women, and effectiveness was used as the criterion to answer the second question, which is whether the active policies approved so far have had a significant impact on the youth labour market in the agricultural sector and especially on the incorporation of women into farm management.

In summary, the profile of the owner and the farm in our case study revealed the effectiveness of the support given to young farmers to set up an agricultural holding in boosting young male and female employability, as it offers a stable, sustainable future for young people in general, and for young women in particular.

## 3. Legislative Developments in Public Policies within the Framework of Gender Mainstreaming in Farming

The current statutory framework for start-up support for young farmers, which underpins our practical case study in this article, is set out in Regulation (EU) 1305/2013 on Support for Rural Development [14], which governs the 2014–2022 programming period for rural development in Europe [15,16].

In theory, the evolution of public policies that promote generational renewal in farming and encourage the incorporation of women to the agricultural sector in order to ensure its sustainability and survival are inevitably linked to the evolution of the CAP [17,18], and more specifically, the second cornerstone, which refers to rural development policy.

The CAP is recognised as one of the European Union's most important policies, since it is common to all EU Member States and is administered and financed at European level. It receives a large part of the EU budget, as well as co-financing from the Member States.

The basic principles that inspired its inception in 1962, following the approval of the Treaty establishing the European Community, included the following: (a) market unity, in order to achieve agricultural price harmonisation and a food safety system in the different Member States; (b) financial solidarity, through the approval of a Community financial instrument known as the European Agricultural Guidance and Guarantee Fund; (c) the principle of Community preference, establishing protective measures with regard to non-Community market members.

In 1992 [19,20], the measures introduced by the MacSharry reform of the CAP had a huge impact on Spanish agricultural policy, giving rise to Law 19/1995, of 4 July 1995, on the Modernisation of Agricultural Holdings (LMAH) [21–24], which is the legislative framework for the current rural development policy in Spain.

Section IV of the LMAH is devoted entirely to granting financial aid to young farmers setting up first-time farms as owner or partners on priority agricultural holdings through special tax concessions in order to revitalise farming.

Out of all the European regulations that govern agricultural policy in rural areas, it is important to mention Regulation (EC) 1257/1999 of 17 December 1999 on Support for Rural Development, which led to the creation of the Agenda 2000 [25–31], given that it was the first time that it included funding for young farmers to set up agricultural holdings in its list of action measures. Its relevance has been reflected in all successive European regulations.

In terms of gender, Law 35/2011, of 4 October, on the Shared Ownership of Agricultural Holdings was one of the most important legislative milestones to the incorporation of women into the Spanish rural community.

Given the vulnerability of women's work in agriculture, mainly due to their lack of economic, professional, and social recognition, policymakers decided to promote shared ownership of farms, defined as "that which is made up of a married couple or partners in an affective relationship that jointly manages the farm".

The legislator's intention was to introduce female participation in the management of family farms for the first time, so as to enable women to benefit from all the rights

associated with their work on the farm and also to take on the related managerial decisions and responsibilities.

In fact, this law, in addition to regulating the system and taxation of shared-ownership farms, recognises women's monetary remuneration for their actual involvement in the farm when they have effectively contributed to farming activities without receiving any payment or consideration for the work undertaken.

The 2030 Agenda and the Sustainable Development Goals (SDGs) currently define the action plan in rural areas of the UN member states, including the European Union and Spain, with a view to achieving sustainability in our world, leaving no one behind and benefiting people, the planet and prosperity.

The fifth SDG aims to achieve gender equality and the empowerment of women by drawing up policies to promote female participation in management and leadership bodies in the rural world.

In this sense, the Spanish government is committed to fostering the law on shared ownership and increasing the number of women listed in the Shared Farm Ownership Register and to showcasing the work carried out by women as a cornerstone for rural development.

The other main SDG to be highlighted is number eight, related to promoting entrepreneurial spirit among young people in the rural world in order to ensure populated, prosperous, well-organised, well-connected rural areas. To achieve this, the government is intent on adapting regulations and offering public funding to strengthen the network of rural micro-SMEs and to establish a labour framework for quality employment and a decent work environment in rural settings.

It should be emphasised that the 2030 Agenda and the SDGs are part of the future of European rural development policy which is geared towards the "European Green Deal" [32,33].

The Green Deal is regarded as the European Union's strategy to promote a sustainable, competitive agricultural sector. It intends to tackle this challenge by investing in more environmentally friendly technologies, in cleaner transport systems and in more energy-efficient housing and buildings. The ultimate aim is for the European Union to be climate neutral by 2050.

There is no doubt that the European Union and its Member States are aware of the need to sustain public policies aimed at ensuring agricultural sustainability and competitiveness and developing rural areas. They attach particular importance to measures aimed at meeting the needs of young male and female farmers.

## 4. The Situation of Young Women in Agriculture

The role of rural women is key to territorial and social organisation in the rural environment. However, there are more scenarios of inequality between men and women in agriculture than in urban settings.

As Figure 1 shows, on the one hand, horizontal segregation in the Spanish labour market persists, since men make up the majority of the labour force in all productive sectors except for services. In addition, the percentage of both active male (5.9%) and female (2.7%) farmers compared to other economic sectors is alarmingly low.

Hence, women in the rural environment are subject to double discrimination, firstly, because they are women and secondly, because they live in rural areas.

In addition, if we focus on gender and age bracket in agriculture, as shown in Figure 2, the highest percentage of both working men (67%) and women (71%) are in the 40 to 65 age group. While in the 20 to 40 age bracket, the percentage of employed persons only accounts for 26% of the total workforce in the case of men and 23% in the case of women. This shows the lack of appeal of the rural world and of agriculture for young people and highlights the need for positive legislative measures and proactive policies.

Finally, Figure 3 shows that in the most recent agricultural census, which dates back to 2009, 69.74% of farm owners were men. The graph also highlights the vertical segregation

existing in farming, which is characterised by the scant representation of women in the ownership of agricultural holdings.

Moreover, the highest percentage of farm owners is concentrated in the 55 and over age group. The number of farm owners under 40 years of age is insignificant. Thus, the figure of the young farming entrepreneur is not a major reference.

In short, it can be stated that rural entrepreneurship is eminently masculine, with scarce visibility of female agricultural entrepreneurs. There is also an ageing rural population, both in the case of farm managers and its labour force, highlighting the lack of appeal of the rural world for young people.

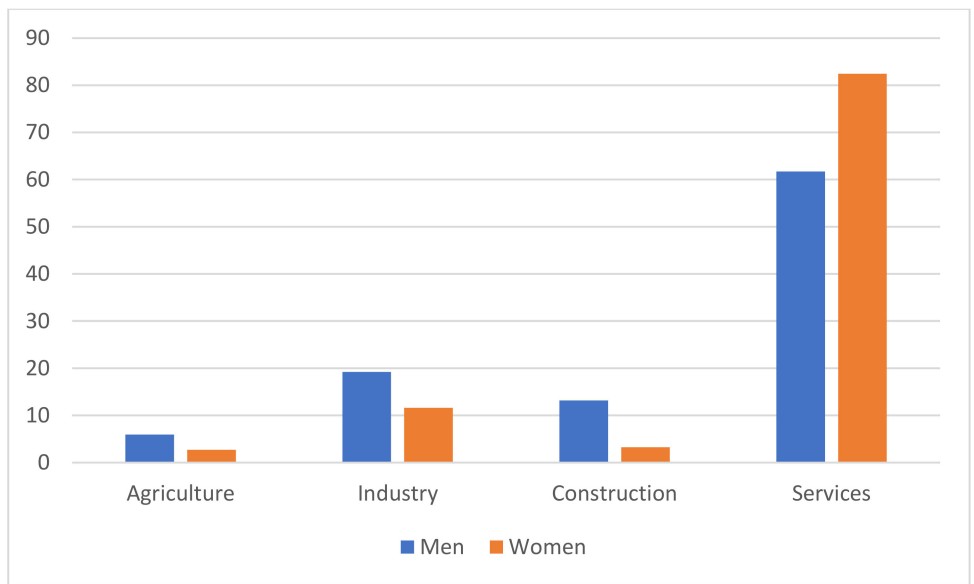

**Figure 1.** Percentage breakdown of the workforce by economic sector and gender in 2020. Source: compiled by the authors based on INE data [34].

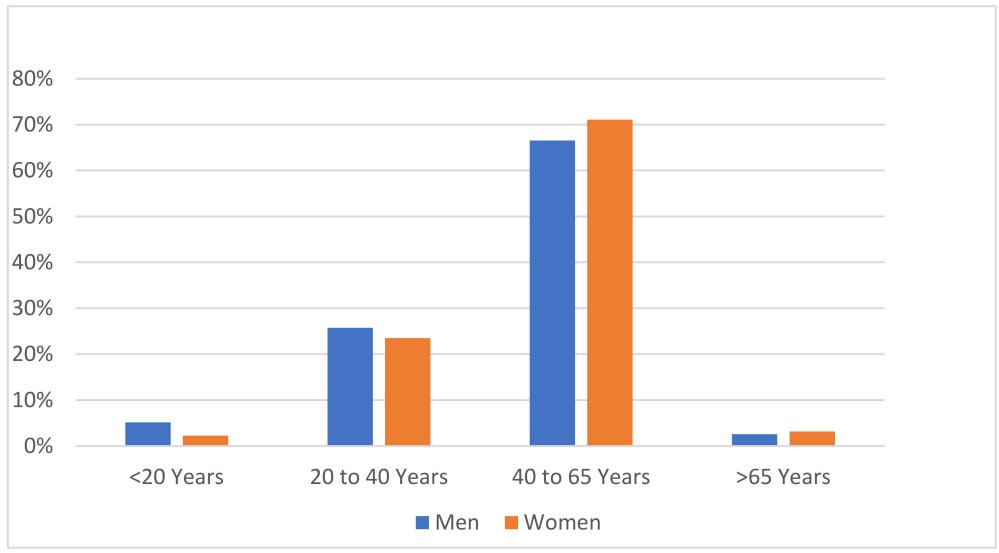

**Figure 2.** Percentage breakdown of the workforce by gender and age in agriculture. Source: compiled by the authors based on INE data [35].

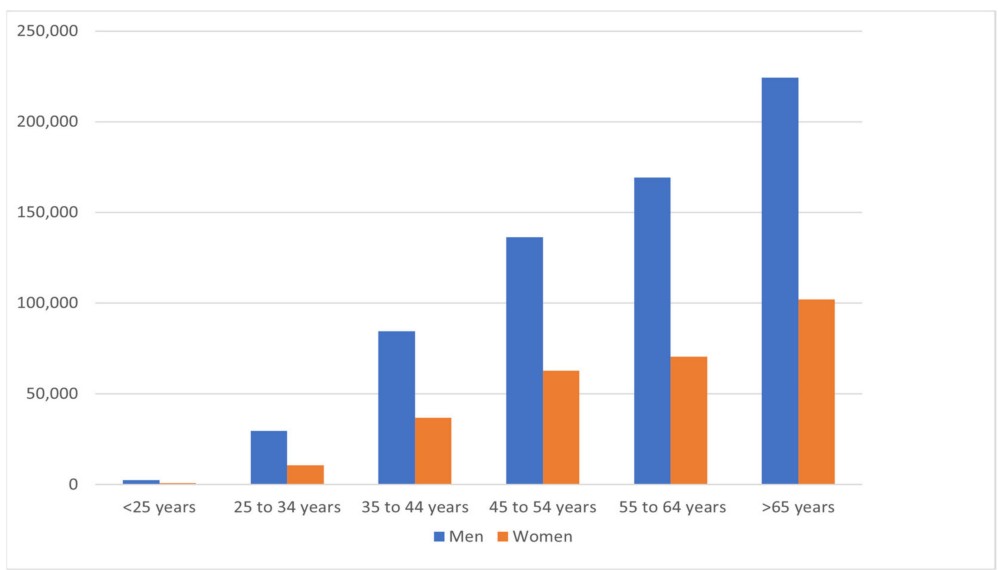

**Figure 3.** Number of farm owners by age and gender in 2009. Source: compiled by the authors based on INE data [36].

## 5. Discussion and Results

To verify the effectiveness of the measures aimed at supporting young farmers in setting up their first farm, as established in Regulation 1305/2013, we performed a case study of a young female farmer setting up a farm for the first time to assess whether they are a viable alternative to providing a decent, stable livelihood for young women.

### 5.1. Legislative Framework for Supporting Young Farmers in Setting Up Their First Farm

First of all, we shall proceed to briefly analyse the legislative framework of the measures aimed at supporting young farmers in setting up their first farm, as set out in Regulation 1305/2013 on rural development policy. To do so, we first need to examine European legislation and then move on to local legislation, in this case Valencian legislation, as this is the region where our case study of a young farmer setting up their first-time farm is based.

In the European framework, funding is included in the measure laid down in Article 19 of Regulation 1305/2013 on Support for Rural Development, which refers to the development of agricultural holdings and businesses.

Specifically, according to Article 19, firstly, funding is earmarked for setting up businesses, both for young farmers and for non-agricultural activities in rural areas, as well as for developing small agricultural holdings, all subject to the drawing up of a business plan. Secondly, funding is for investments in setting up and developing non-agricultural activities, with support being granted to micro and small enterprises, people in rural areas, and farmers or members of a farming household. Lastly, it includes the provision of an annual payment or a one-off payment to farmers who are eligible for the small farmers scheme established in Title V of Regulation 1307/2013 on direct payments to farmers under support schemes included in the CAP framework and who permanently transfer their holding to other farmers.

It should be noted that this measure is especially relevant to achieve several of the six priorities set out in Regulation 1305/2013. In fact, it makes a direct, essential contribution to the sixth priority on promoting the economic development of rural areas, reducing poverty, and fostering social inclusion, as well as to the second priority on diversifying into non-agricultural activities that help to promote the development of non-agricultural activities that have a positive impact on economic development in rural areas.

Within the framework for implementing support for young farmers under Article 19 of the Regulation, in the case of Spain and thanks to the broader autonomy that the

Regulation establishes for Member States to develop their own rural development policy, a programming structure based on a National Rural Development Framework (NRDF) [37], a specific programme for the National Rural Network and 17 Regional Rural Development Programmes (RDPs), and a rural development programme for each of the Spanish Autonomous Regions was created.

The regional programme strategy must respond to regional criteria and must be consistent with the national programme, taking into account the common aspects established in the national framework.

It is clear from both the National Framework and the 17 RDPs that funding for the initial set up and participation of women in the rural world is one of their core objectives, since one of the needs identified in the SWOT analysis of the different programmes refers to safeguarding stable employment in the rural environment, especially for young people and women, through the creation of small and medium-sized enterprises by these sectors of the population.

Lastly, we shall examine this measure in the Valencian Region's RDP [38]. Specifically, sub-measure 6.1 of the Valencian RDP on funding for young farmers to set up in business, which is part of measure 6 to develop farms and businesses, is considered essential to provide young people with a good working future and a decent livelihood by helping them to set up viable, sustainable and competitive agricultural holdings.

The aim is to encourage young people to move from other productive sectors to farming in order to generate stable, sustainable employment for the young population.

It should be emphasised that the aim of the funding for young male and female farmers to set up business is to improve the viability of agricultural holdings, to reorganise and modernise the farming industry and to promote generational renewal.

This creates the best possible conditions for young, well-trained, highly skilled and enthusiastic people to move into farming. This has a direct impact on bettering the economic performance of farms by introducing new management methods.

*5.2. Requirements for Obtaining Funding*

We shall begin by analysing the requirements for applicants and the holdings that must be met by possible beneficiaries of the measure in order to subsequently check that the farm in our case study is suitable and eligible for the funding for young farmers laid down in Article 3 of Order 7/2015 of 1 December, which sets out the conditions that must be complied with for these subsidies.

The criteria relating to the applicant, i.e., the owner of the holding, require him/her to be a young farmer. This means they must be between 18 and 40 years of age at the time of applying, have sufficient professional agricultural qualifications and skills, and be setting up a farm for the first time as a tenant or owner farmer.

In addition to the age requirement, the farmer must be a working farmer and use a government-recognised advisory service. To be considered as a professional farmer, at least 50% of their total income must come from agricultural or other related activities. The proportion of income obtained directly from farming on their holding must not be less than 25% of their total income.

Finally, it should be noted that the owner of the farm is required to comply with other general requirements for access to any type of public subsidies, such as (a) compliance with the regulations for the integration of disabled people in the workplace, (b) not being subject to any of the conditions set out in the General Law 38/2003 on Subsidies, (c) proof that they are up to date with their tax and social security obligations, and (d) registration as an agricultural entrepreneur with the Tax Agency and in the corresponding social security regime.

We shall now move onto the requirements that refer to the agricultural holding.

The farm must be classified as a priority farm and comply with the necessary requirements in each case. These include the agricultural income obtained from the farm being equal to or greater than 35% and less than 120% of the reference income. The process of

actually setting up the farm must have begun within a maximum period of 24 months prior to the date on which the funding application was submitted. In addition, the actual start-up must take place within a maximum of nine months after the date on which the subsidy is granted.

This list of requirements is subject to the presentation of a business plan for a first-time start-up as owners or partners of priority shared holdings, provided that the young farmer is in control and is responsible for the civil, tax, and social aspects of the farm's management. It must describe the objectives, phases and agricultural activities to be carried out on the new holding.

Last but not least, the final requirement is to comply with the provisions of Decree 3/2006, of 10 March, of the Valencian Regional Government, which establishes the regulations for the implementation of Law 2/1989, of 3 March, of the Valencian Regional Government, on environmental impact, to determine the requirement for the environmental impact statement or estimate [39].

Therefore, we can state that the farm we have studied and described comprehensively in the methodology section meets the requirements set out in the application for funding for first-time farms by young people set out in Order 7/2015 of the Valencian Regional Ministry of Agriculture.

*5.3. Technical and Economic Study of the Farm*

The following tables summarise the farm's overheads and its viability according to the indicators established in Order 7/2015 of the Valencian Regional Government. In the case of the annual overheads, the measure used refers to the AWU.

In our case, the agricultural use is to grow stone fruit with irrigation, and based on this, an AWU of 0.213 per hectare was established. Therefore, if we multiply this figure by our farm's 11.61 hectares, we obtain 2.47 AWUs. This enables us to easily calculate overheads.

Table 1 shows the cost of the overheads involved in starting up the agricultural holding. They are broken down into personnel costs, rent, risk insurance costs, depreciation, and maintenance costs of fixed assets and other overheads.

**Table 1.** Overheads.

| Summary of Overheads | Amount/AWU | Amount (EUR) |
|---|---|---|
| Agricultural social security | 1593.18 | 3935.15 |
| Operating insurance | 790.00 | 1951.30 |
| Marketing and operating costs | 152.38 | 376.38 |
| Operating leases and rentals | 5611.96 | 13,861.54 |
| Other overheads | - | 3035.00 |
| Annual overheads | | 23,159.37 |
| Salaries | 13,620.60 | 13,620.60 |
| | % Expenditure | Amount (EUR) |
| Machinery depreciation | 12.5 | 6500.00 |
| Maintenance and machinery costs | 0.05 | 2600.00 |
| **Total overheads** | | **45,879.97** |

Source: compiled by the authors.

With regard to personnel costs, it is important to point out that the labour provided by the owner of the farm is equal to the maximum of 1 AWU, which is equivalent to 1920 h per year, the maximum number of working hours permitted under Spanish labour legislation.

Therefore, in addition to the farm owner, salaried labour will be hired to the value of one AWU and the remaining 0.47 AWU will be supplied by a temporary employment agency.

The personnel costs for 1 AWU are EUR 13,620.60, as shown in Table 1, plus the agricultural social security costs for the 2.47 AWUs, which amounts to EUR 3935.15.

The tangible fixed assets include a Deere tractor with a purchase price of EUR 52,000.

Therefore, applying the tax depreciation tables considered in the Order that regulates this subsidy and refer to the purchasing value updated in the year in which the business plan is completed, we obtain the depreciation and maintenance costs of EUR 6500 and EUR 2600, respectively, as shown in the table above.

The heading "Other overheads" in Table 1 includes the following expenses that must be made for the start-up of the farm:

(a)  The integrated pest management consultancy contract required for the farm in accordance with Royal Decree 1311/2012 of 14 September, which establishes the framework for action to achieve sustainable use of plant protection products [40]. This has a market price of EUR 1000.

(b)  The EUR 60 annual fee for the Valencian Agriculture Association's (AVA-ASAJA) technical and legal advisory service.

(c)  The EUR 200 fee for website maintenance services.

(d)  The EUR 200 annual fee for membership of the Xativa Cooperative.

(e)  The cost of contracting service companies to provide qualified personnel for the most complex tasks in the field. The average cost of the service per AWU is EUR 2500. In our case, we have 0.47 AWUs subcontracted through a temporary employment agency. This is equivalent to EUR 1175.

(f)  The EUR 430 fee for organic production control and certification services charged by the Valencian Region's Organic Agriculture Committee (CAECV).

Table 2 shows the calculations required to assess the economic viability of the farm. First of all, the agricultural income per AWU must be calculated after the implementation of the business plan in order to compare it to the reference income published annually, which for 2021 stands at EUR 30,622.23.

**Table 2.** Economic viability of the farm.

| Indicator | Indicator Definition | Amount after the Business Plan |
|---|---|---|
| Gross margin | 24,886.33 AWU (Order 7/2015) | EUR 61,469.24 |
| Net margin | Total gross margin−Total overheads | EUR 15,589.27 |
| Salaries paid | Total salaried workforce | EUR 13,620.60 |
| Reference Income | 30,622.23 for 2021 (Official State Gazette 18/12/2020) | EUR 30,622.23 |
| No. AWUs in the holding | 0.213 AWU/Hectare (Order 7/2015) | 2.47 |
| No. AWUs of labour force | Tables in Annex I applicable to Order 7/2015 | 2 |
| Agricultural Income | (Net margin + salaries paid) No. AWUs of labour force | EUR 14,604.94 |
| Indicator for agricultural income over reference income | $\frac{\text{(Agricultural Income)} \times 100}{\text{Reference Income}}$ | 47.69% |

Source: compiled by the authors.

The agricultural income is obtained by adding the net margin of the farm (EUR 61,469.24−EUR 45,879.97) and the salaries paid (EUR 13,620.60) divided by the number of labour AWUs (2 AWUs, corresponding to 1 AWU of the farm owner plus 1 AWU of salaried labour).

In accordance with the guidelines laid down by the Valencian Regional Ministry of Agriculture (Order 7/2015), after the completion of the business plan, the viability of the farm project submitted must be guaranteed. To this end, the agricultural income from the agricultural holding after the implementation of the business plan must be between 35% and 120% of the 2021 reference income of EUR 30,622.23.

In other words, the economic return generated in the agricultural holding that is attributed to the work unit must represent between 35% and 120% of the reference income, i.e., between EUR 10,717.78 and EUR 36,746.68.

Based on the data shown in Tables 1 and 2, in our case, the agricultural income per AWU is 47.69% of the reference income. Therefore, we can state categorically that our case study farm is completely viable from an economic point of view.

### 5.4. Scoring Criteria and Amount of Funding

The criteria set out in Order 7/2015 of the Valencian Regional Ministry of Agriculture for the evaluation of applications to grant funding to young farmers to set up a first-time farm, as well as the points given to these criteria that determine the total amount of funding granted to our case study farm are shown below.

Table 3 below shows that the application presented in this paper would obtain a score of 30 points according to the criteria established by the Valencian Regional Government. A minimum of 12 points is required for applications to be approved and in our case, this is clearly exceeded.

**Table 3.** Scoring criteria for young farmer applications for first-time farm set up.

| Criteria | | Score |
|---|---|---|
| Applications where the business plan envisages the use of more than 2 AWUs on the holding after the implementation of the business plan. | X | 10 |
| Applications from young people setting up a farm by joining associative entities that qualify as priority holdings where the farm is controlled by the beneficiary of the funding. | | 10 |
| Applications relating to holdings located in areas with natural or other specific constraints as referred to in Article 32 of Regulation 1305/2013 or included in the Natura 2000 network. | | 5 |
| Farms located in municipalities covered by the LEADER measure not included in the previous point. | | 3 |
| Applications from women who aim to take over the ownership of a farm, either individually or through shared ownership as defined in Law 35/2011 on the Shared Ownership of Holdings. | X | 5 |
| Applications made under shared ownership. | | 5 |
| Applications that include actions that have an environmental, energy-saving, or climate change objectives. | X | 5 |
| Applications where more than 50% of the AWUs of the holding are dedicated to organic production or other quality production schemes as described in Article 16(1)(a) of Regulation (EU) 1305/2013. | X | 7 |
| Applications that include the farm holder joining a cooperative for processing and/or marketing at least 25% of the holding's production. | X | 3 |
| Applications that comply with the sectorial and productive orientation guidelines established by the competent Regional Ministry in each call for applications. | | 10 |
| TOTAL | | 30 |

Source: compiled by the authors.

Firstly, as we have stated throughout this article, women are underrepresented in farming, as shown in Figure 1 by the number of farm owners. In addition, the latest agricultural census of 2009 shows that only 30.26% of all farms were owned by women.

In this sense, and in order to encourage women to become farm managers, there is a criterion that awards five points to the application if the farm is owned by women through shared or individual ownership, as is our case.

Therefore, as we have argued in the introduction, there is positive gender discrimination in the evaluation of applications for start-up funding for young female farmers.

Other noteworthy aspects are, firstly, the points awarded for setting up a farm that has over 2 AWUs after the implementation of the business plan. In our case, the farm has 2.47 AWUs, thus giving us an additional 10 points in the evaluation.

Furthermore, as our farm intends to engage 100% in organic production, it receives an additional score of seven points.

Previously, we mentioned that one of the sales channels would be through membership of a cooperative (Xativa Cooperative) for the processing and marketing of at least 25% of the farm's production. This gives the farm an extra three points.

Lastly, applications that include actions that involve an environmental objective of saving energy or combating climate change receive an additional five points. Our case complies with this criterion as it features production methods that do not use nitrogenous chemical fertilisers and are based on renewable energies that reduce the consumption of $CO_2$ emissions.

Table 4 shows the funding amounts per criterion and the total amount of funding that our case study farm would receive. This figure would be EUR 30,000.

**Table 4.** Funding amounts for young farmers to set up a first-time farm.

| Criterion | Amount | Funding Requested |
|---|---|---|
| Minimum subsidy | EUR 20,000 | EUR 20,000 |
| The young farmer sets up business as an associative entity, other than sole ownership, with legal status, owning an agricultural holding that qualifies as a priority holding after the implementation of the business plan. | EUR 5000 | |
| The farm expects to use more than 2 AWUs and up to 3 AWUs after the implementation of the business plan. | EUR 10,000 | EUR 10,000 |
| The farm expects to use more than 3 AWUs after the implementation of the business plan. | EUR 15,000 | |
| The farm is mixed, with at least 0.5 AWUs for all agricultural activities and 0.5 AWUs for all livestock activities after the implementation of the business plan. | EUR 10,000 | |
| The majority of the holding is located in areas with natural or other specific limitations, according to the Valencian Region's 2014–2020 RDP or it is located in areas included in the Natura 2000 network. | EUR 5000 | |
| The farm is located mostly in a mountainous area, according to the Valencian Region's 2014–2020 RDP. This criterion is not compatible with the previous one. | EUR 15,000 | |
| The holding has facilities for the packing and/or processing of at least 25% of the production obtained from the holding. | EUR 5000 | |
| The business plan provides for the recovery of abandoned land amounting to at least 0.3 AWUs. | EUR 5000 | |
| TOTAL | EUR 90,000 | EUR 30,000 |

Source: compiled by the authors.

It is important to point out that simply by engaging in agriculture as a young farmer and complying with a series of requirements that we have described in this article, we receive EUR 20,000 to defray the start-up costs of the farm. Furthermore, in our case, we would receive an additional EUR 10,000 because our case study farm envisages using between 2 and 3 AWUs after the implementation of the business plan.

In short, the amount of this subsidy represents 65.38% of the costs involved in starting up our farm, given that the total overheads come to EUR 45,879.97 and external financing amounts to EUR 30,000. This clearly demonstrates our theory on public protection of the sector and the consideration of farming as a viable alternative for the future employment of young entrepreneurs and in particular for young women.

*5.5. Evolution of Beneficiaries of First-Time Farm Set Up Funding in The Valencian Region*

The official data on the number of beneficiaries of first-time farm set up funding in the different calls published by the Valencian Government under Order 7/2015 (Figure 4) show that although there was a small increase in the total number of applications approved in the 2017 and 2019 calls compared to the 2016 call, the number of female beneficiary applications has not changed considerably and actually fell slightly in 2019.

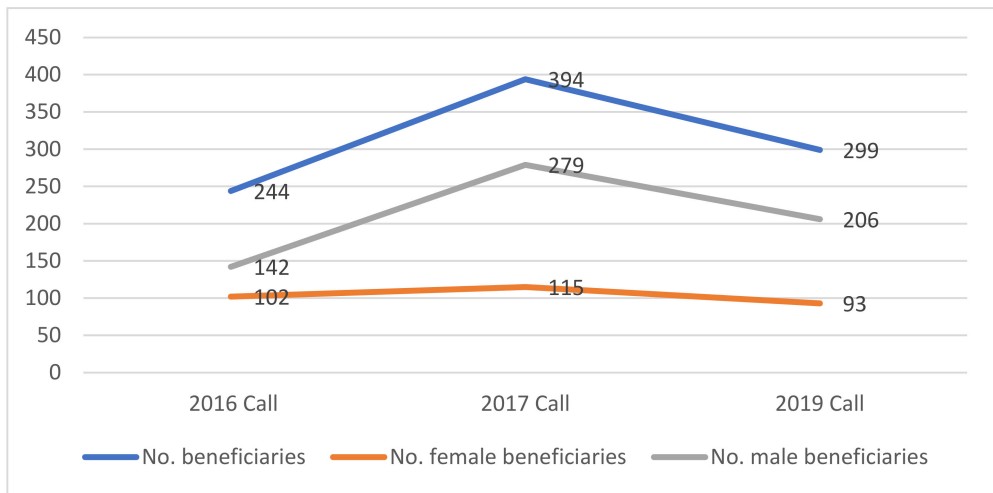

**Figure 4.** Number of beneficiaries of the measure by call and gender. Source: compiled by the authors based on Valencian Government data [41].

Although it is true that we do not have official data on the number of applications submitted, in the light of the results of Figure 5 shown above, we can confirm the general ineffectiveness in the Valencian Region of the funding for young people to set up their first farm. This is especially significant in the case of young female farmers.

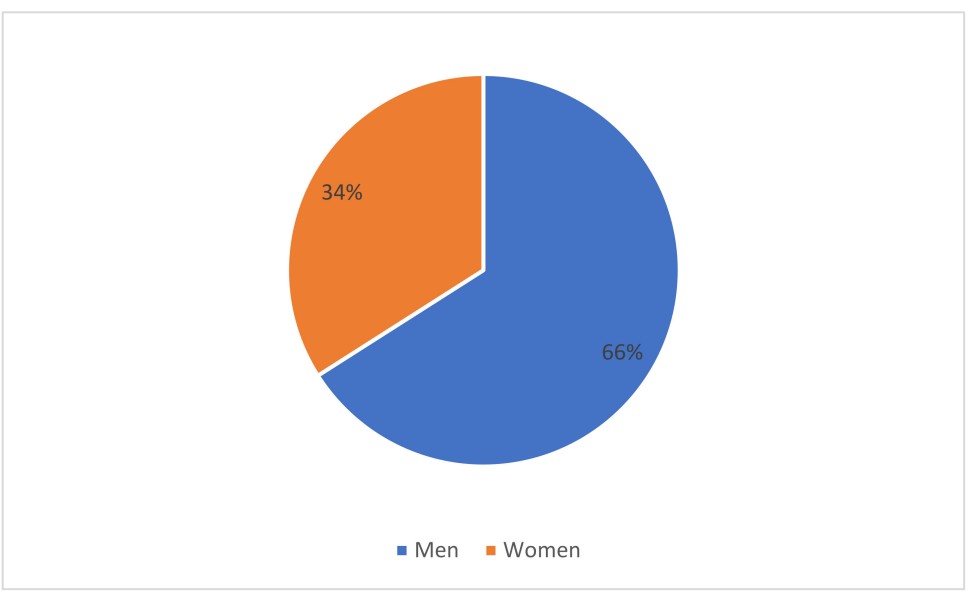

**Figure 5.** Average number of beneficiaries in the different calls by gender. Source: compiled by the authors based on Valencian Government data [42].

Indeed, the low number of female beneficiaries of these subsidies shows the poor participation of women in the management of agricultural holdings as owners, given that, even though there is positive discrimination in the ranking of applications in favour of women, the number of female beneficiaries represents an average of 34 percent of the total number of beneficiaries in the three calls for applications put out by the government.

The fundamental factor affecting the low percentage of female beneficiaries is the difficulty in accessing farm ownership and management as established in the Spanish Shared Ownership Law of 2011, which seems to have failed for the time being. In addition to legal solutions, cultural and social changes are required, as in the rural world there are

many stereotypes and gender roles that influence the poor participation of women in farm management and their vulnerability in a masculine environment [43].

## 6. Conclusions

This article demonstrates that there is clear governmental protection for farming, with European legislation on agriculture, the LMAH, the Law on the Shared Ownership of Agricultural Holdings, as well as the new world map on rural issues established in the 2030 Agenda and the Sustainable Development Goals, giving priority to measures for young farmers to set up agricultural holdings and to improve female incorporation to the rural world.

In particular, it is worth highlighting the fifth SDG, which is to achieve gender equality in the rural world, empowering rural women and designing policies for the participation of women in the management and decision-making of farms.

The other relevant SDG is Goal 8 which centres on promoting youth entrepreneurship in rural areas by encouraging young people to start businesses.

All these public policies aimed at helping young people and women to set up business in the rural world centre on the modernisation and future sustainability of the agricultural sector for various reasons, including the transfer, innovation, and development of rural know-how, the improvement of farm viability and competitiveness, and the creation of small and medium-sized enterprises that are directly and indirectly related to farming and have an impact on the maintenance of stable employment in rural areas, especially for young people and women.

After analysing the legal framework of the positive measures designed to encourage young farmers to set up business in Europe, in Spain, and in the Valencian Region's RDP, we sought to check whether they were effective by testing their implementation through a case study of a young female farmer setting up her first farm. The case study included the requirements for accessing funding, a technical and economic study of the farm's viability, and the scoring criteria and amount of funding granted.

The profile of the owner of our case study farm is a woman between 18 and 40 years of age, with a university degree, who has the required professional qualifications to apply for start-up funding and who is going to set up for the first time as a sole owner on a priority farm.

As a result of the analysis of the technical and economic study of our case study farm we can conclude that;

Firstly, our case study farm is economically viable, since the agricultural income per AWU represents 47.69% of the reference income and is therefore within the range of 35% and 120% established in the guidelines set out in Order 7/2015 of the Valencian Regional Ministry of Agriculture and our case study farm complies with the aforementioned requirements on applying for start-up funding for young farmers.

Secondly, the amount of funding granted in our case represents 65.38% of our total start-up costs. This represents sufficient external financing to be able to set up a farm for the first time without taking on considerable financial risk.

Lastly, positive discrimination in favour of young female farmers has been demonstrated. A decisive effort is being made to encourage young women to join the rural world by giving them greater priority in accessing funding for young farmers to set up agricultural holdings. In other words, the aim of the public authorities is to boost the role of rural women, which is fundamental for the territorial and social organisation of the rural world.

In short, the efficiency of these subsidies for young people to set up farms in terms of employment opportunities for young women in agriculture has been conclusively demonstrated in the study of our farm.

However, as we have seen in this research, firstly the percentage of young men and women working in agriculture (the age range defined for "young people" is deemed to be between 18 and 40 years of age according to legislation) was still low in 2020, accounting

for around 24% of the total number of people active in the agricultural sector. Secondly, the persistence of horizontal and vertical segregation in the Spanish labour market and the agricultural sector has been confirmed, as well as the low number of female beneficiaries for young people to set up their first farm in the different calls for applications put out by the Valencian Regional Government.

Therefore, with regard to the second question raised, we can affirm the ineffectiveness in Spain, and specifically in the Valencian Region, of funding for young farmers, as well as the inefficiency of active European and Spanish public policies to encourage young men and women to set up farms. The results of the measures implemented over 20 years ago have not managed to change the structure of the Spanish workforce, which is mainly employed in the service industry and to a lesser extent in agriculture. In addition, they have not improved the visibility of the figure of female agricultural entrepreneurs and have not increased the number of young women accessing this funding in the Spanish case.

In this sense, neither the European regulations listed throughout the article, nor the LMAH or the Law 35/2011 on the Shared Ownership of Agricultural Holdings have been sufficient in Spain and in the Valencian Region. They have therefore failed to achieve the incorporation of young men and women into the Spanish agricultural sector. It remains to be seen what impact the 2030 Agenda and the SDGs, as well as the new CAP programming period, will have on the agricultural sector.

Promoting "rural" activities as a stable and sustainable way of life is essential to attract young people and encourage them to move from other productive sectors to farming, thus fostering youth employment.

The future challenges for this type of funding will be for government to make these initiatives more widely known and more appealing and make it easier for young and female farmers to gain access to the tenure of rural holdings, whether they are owned or rented, as this is currently a very complex issue.

As a final reflection, in addition to the legal solution of active policies that promote generational renewal and the incorporation of women into the agricultural sector, cultural and social changes are necessary, given that the presence of stereotypes and gender roles show that the patriarchal system is still very present in the rural population.

**Author Contributions:** Conceptualization, B.L.G., I.P.-J. wrote the first draft; B.L.G., I.P.-J. revised the paper and conceptualized the subtitles about single cell protein substrates; B.L.G., I.P.-J. revised the draft and the final form; B.L.G., I.P.-J. helped in gathering data and discussions. Both authors have read and agreed to the published version of the manuscript.

**Funding:** This research received no external funding.

**Institutional Review Board Statement:** Not applicable.

**Informed Consent Statement:** Not applicable.

**Data Availability Statement:** Not applicable.

**Conflicts of Interest:** The authors declare no conflict of interest.

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
