# Peer review of "Are Public Subsidies to Encourage Young Farmers Effective? Case Study of a First-Time Farm Set Up by a Young Female Farmer in the Valencian Region of Spain"

_sustainability, doi:10.3390/su13169320_

Round 1

Reviewer 1 Report

The paper is on an interesting issue of farm renewal and supporting entry to farming by women and young people in general. The overview of the legal framework is good. The case evidence is relevant but needs to be better motivated and linked to the legal framework discussion. The methodology section is quite unclear and represents a mix of diffrent things comprising the paper. More specifically, I see the so called simulation as a case study of one farm (or a model farm). Therefore, a case study methodology would be appropriate to be introduced. Then, it is necessary to motivate the particulat farm choice and thus establish validity of the study. The conclusions should be better linked to the results of the case study and some more relevant discussion will be useful.

Author Response

Thank you very much for your very constructive comments.

In the revised article, we have changed the methodology section in line with your criteria for a case study methodology.

To do so, we have described the profile of the owner and the farm studied, justifying  the choice.

The chosen profile of the farm owner meets the two requirements of being young and female, which enables us to answer the question posed about the fact that support for first-time set-up offers a decent, stable working future for young people in general, and women in particular

The farm presented in this case study is located in the municipality of Xativa, in the province of Valencia. The geographical choice of the farm is basically due to two reasons:

Firstly, the municipality of Xativa is ideal for planting irrigated stone fruit crops, thus optimising resources. Secondly, we are linked to the municipality of Xativa because of an agricultural family tradition and because it is an area characterised by agricultural masculinization.

Also, We used efficiency and effectiveness as the criteria to answer the two questions posed. Efficiency was the criterion used to answer the first question, which is whether public subsidies for first-time farm set-up contribute to the employability of young men and women, and effectiveness was used as the criterion to answer the second question, which is whether the active policies approved so far have had a significant impact on the youth labour market in the agricultural sector and especially on the incorporation of women into farm management.

And finally, we have added a section in the article relating to the evolution of the number of aid beneficiaries in the different calls for applications under Order 7/2015 of the Valencian government to determine the effectiveness of public policies that promote generational change and the incorporation of women.

Thank you

Reviewer 2 Report

Dear authors.

Congratulations for your work on the impact of public subsidies that aim to encourage young farmers (especially young female farmers) to set up a new agricultural business in rural areas. The research problem is really interesting and important in order to achieve real sustainability in rural areas. However the following issues need to be addressed in order to better respond to the set objectives.

  1. The research design. The objective seems to be an assessment of the effectiveness of those public programs. However, study is based only on document review and a simulation. This shows that the policies are addressing the matter, but not their results. Therefore, the research question remains partially unanswered. It would have been interesting to analyze the Selection Reports to see if there is a statistical significant increase in the number of young female farmers that were supported by the measure. The matter is indeed acknowledged as a limitation (lines 557-567), however in this case:
  2. the results should be better discussed. This section should more clearly emphasize the study contribution to the current theory and research on the analysed topic. Were similar results reported in other areas? What lessons they offer (especially in the eve of a new EU CAP Programming Period) ?
  3. Could the results of a single simulation be generalized for other areas?

Some minor comments:

  • in the lines 273-278 is stated that the second (2nd) EU rural development priority (from the Reg. 1305/2013) deals with the diversification of economic activity by promoting non-agricultural activities. However the priority deals with "enhancing farm viability and competitiveness of all types of agriculture in all regions and promoting innovative farm technologies and the sustainable management of forests,  with a focus on the following areas:" (Reg. 1305/2013).
  • in the simulation the project receives 30 points (Table 3, lines 461-463). Is there a minimum amount needed to be considered for selection? Compared to the previous calls of the measures, are 30 points enough to be selected? (This is by no means a certain indicator, but it builds the context
  • This point is only for my own curiosity. Is there a similar situation in the case of the measures of Local Action Groups? Could they be a more favorable option? Since they function under the community-led local development (CLLD) principles, which are based on social participation and inclusion.

Author Response

Thank you very much for your very constructive comments and for encouraging us to continue our research.

We have followed your criteria , We used efficiency and effectiveness as the criteria to answer the two questions posed. Efficiency was the criterion used to answer the first question, which is whether public subsidies for first-time farm set-up contribute to the employability of young men and women, and effectiveness was used as the criterion to answer the second question, which is whether the active policies approved so far have had a significant impact on the youth labour market in the agricultural sector and especially on the incorporation of women into farm management.

To answer the second question, we have included a section entitled the number of beneficiaries of first-time farm set-up funding in the different calls published by the Valencian Government under Order 7/2015 to determine the effectiveness of public policies that promote generational change and the incorporation of women in the agricultural sector. We can conclude by stating that these public policies have failed and that they must be reconsidered in the new CAP programming period.

These public policies that encourage generational change and the incorporation of women in the agricultural sector could be applied to other economic sectors such as construction because, as we have seen in this article, it is a sector characterised by the scarce presence of women. Although it is true that the public authorities do not grant public protection to the construction sector as they do in agriculture.

In addition, it should be noted that we have included in the line 475 The minimum score that applications for aid for the first installation must have in order to be approved (12 points). (12 points). And we have also modified the methodology secctión used a case study methodology.

Thank you very much for your comments

Round 2

Reviewer 1 Report

I am happy with the revision. Please, do a final round of proofreading.

Reviewer 2 Report

Dear authors,

The changes you have made highly increased the quality of the study. For future works please try to put more emphasize on the contribution of the articles to the literature and the research subjects that it opens. 

All the best!